# Botulinum Toxin Treatment in Hereditary Spastic Paraplegia—A Comprehensive Review and Update

**DOI:** 10.3390/toxins17100503

**Published:** 2025-10-13

**Authors:** Bahman Jabbari, Samira Comtesse, Fattaneh Tavassoli

**Affiliations:** 1Department of Neurology, Yale University, New Haven, CT 06510, USA; fattaneh.tavassoli@yale.edu; 2Faculty of Medicine, Goethe University, 60590 Frankfurt am Main, Germany

**Keywords:** botulinum toxin, botulinum neurotoxin, hereditary spastic paraplegia, spasticity, onabotulinumtoxinA, abobotulinumtoxinA, incobotulinumtoxinA

## Abstract

Using Medline and Scopus as search engines, we identified reports of 10 clinical studies (published up to 1 September 2025) on botulinum neurotoxin therapy for hereditary spastic paraplegia (HSP). Nine studies were conducted in adults and one in children. Only one of the ten studies was double-blind and placebo-controlled. The search strategy included only articles published in English and articles providing basic information such as the type of the study, type and dose of the toxin and results of the treatment. Articles not in English, case reports and review articles were excluded. A total of 258 patients were included across all studies. The injected toxin in the open-label studies was botulinumtoxin-A (Botox or Dysport or Xeomin), whereas in the blinded study, the investigators used Prosigne. All open-label studies, which used FDA approved botulinumtoxin-A neurotoxins, demonstrated a degree of motor and non-motor improvement, whereas treatment with Prosigne did not improve patients’ function. The possible reasons for this discrepancy between the blinded study and the open-label studies are discussed. We found no studies on the effect of BoNTs on bladder dysfunction in HSP. There is a need for double-blind, placebo-controlled studies assessing the efficacy of FDA-approved botulinum neurotoxins in children and adults affected by hereditary spastic paraparesis. Such studies should also investigate the effect(s) of early botulinum neurotoxin therapy in this disorder. The novelty of this review is that it represents a comprehensive and critical literature review on this subject, with no other studies of this kind published previously. It also includes data not present in previous reviews of this subject.

## 1. Introduction

Hereditary spastic paraplegia (HSP) comprises a group of inherited neurological disorders characterized by progressive degeneration of corticospinal tract axons, leading to spasticity and weakness in the lower limbs [1]. While spastic paraplegia and urinary dysfunction are the most frequently observed clinical features, HSP is considered one of the most clinically and genetically heterogeneous neurological disorders, encompassing a wide spectrum of symptoms and genetic alterations [2,3]. To date, 87 distinct forms of HSP have been described, with 73 causative genes identified [1]. Given this heterogeneity, classification of HSP is based not only on the pattern of inheritance, but also on the clinical phenotype and molecular pathophysiological mechanisms. Inheritance patterns include autosomal dominant, autosomal recessive, X-linked and, in rare cases, mitochondrial transmission. Clinically, HSP is categorized into two main forms: pure and complex. The pure form is characterized by slowly progressive spasticity and weakness of the lower limbs, signs of corticospinal tract involvement, impaired vibration sense and proprioception, and variably present hypertonic bladder dysfunction. In contrast, the complex form includes the core features of spastic paraparesis along with additional neurological or systemic manifestations such as ataxia, thinning of the corpus callosum, extrapyramidal signs, chorioretinal dystrophy, peripheral neuropathy, and cognitive impairment. Complex forms are more frequently associated with autosomal recessive inheritance than with autosomal dominant transmission [4]. HSP affects individuals across a broad range of ethnicities, with reported prevalence estimates ranging from 1.2 to 9.6 per 100,000 people [5]. Recently, Gan and co-workers’ [6] study on neuroscience and molecular genetics provided a broader mechanistic understanding of neurogenetic disorders such as HSP. Furthermore, Zhu et al. [7] have shown how bioinformatics can be used in understanding complex gene–phenotype relationships in neurological disorders.

HSP is generally characterized by a slowly progressing course over many years, primarily impacting the lower extremities with spasticity and muscle weakness. The age of onset for symptoms is highly variable, encompassing early childhood to late adulthood, and often predicts both the severity and pace of disease progression. Early-onset cases tend to exhibit slower deterioration and retain mobility for longer periods, while adult-onset forms are typically characterized by more rapid functional decline over time [5]. Sex appears to influence both the prevalence and clinical severity of specific HSP subtypes, particularly those associated with SPG4, SPG7, and SPG11 mutations, with some studies reporting earlier onset, faster progression, or greater severity in male patients. However, this association is not consistently observed across all study populations, as some cohorts report no significant sex-based differences in disease onset, progression, or penetrance [8,9].

At present, no curative or disease-modifying therapy exists for HSP, leaving current therapeutic strategies focused on symptomatic relief and functional support [4]. Clinical management typically relies on a multidisciplinary approach aimed at alleviating spasticity, improving mobility, and addressing secondary complications. Pharmacological interventions include oral antispasmodic agents such as baclofen, diazepam, tizanidine, and dantrolene, while intrathecal administration of baclofen has demonstrated efficacy in cases of pronounced spasticity [4]. Anti-spasticity drugs are usually prescribed in increasing doses over days or weeks. Management of severe spasticity may require consuming large doses of these medications, including up to 20 mg baclofen 4 times daily, up to 36 mg tizanidine daily, up to 300 mg dantrolene daily and up to 80 mg diazepam daily. Unfortunately, such large doses are often associated with disabling side effects such as severe fatigue, drowsiness, dizziness and nausea. Urinary urgency may be managed with oxybutynin, provided that structural or infectious causes have been excluded [5]. In addition, physiotherapy and structured exercise programs play a central role in maintaining muscle strength, joint mobility and cardiovascular function. Orthotic devices, such as ankle–foot orthoses, and gait-phase-dependent peroneal nerve stimulation have proven beneficial in correcting foot drop and enhancing gait stability [5]. Emerging experimental therapies targeting microtubule dynamics, oxidative stress and intracellular transport have shown preliminary promise in preclinical models, though their clinical relevance remains to be established [8]. Genetic testing supports diagnostic accuracy and informs family counseling, although the variable penetrance and expression of HSP require cautious interpretation [4].

The clinical spectrum of HSP includes motor and non-motor disorders. The two major motor disorders of HSP are spastic paraplegia and spastic bladder, whereas common non-motor disorders of HSP include pain in spastic limbs, fatigue, insomnia and depression [8,9]. Intramuscular injection of botulinum neurotoxin reduces spasticity by inhibiting acetylcholine release at the neuromuscular junction [10]. Large-scale randomized clinical trials have shown efficacy of botulinum toxin treatment in common clinical causes of human spasticity, such as post-stroke or post-traumatic spasticity, as well as common forms of childhood spasticity such as cerebral palsy [11,12,13]. Several studies have demonstrated that after botulinum toxin therapy, improvement in spasticity in these conditions is associated with improvement in patients’ quality of life [14,15,16]. Botulinum toxin therapy is also devoid of the above-mentioned disabling side effects of anti-spasticity drugs and unlike anti-spasticity drugs, has very limited drug interactions. This review is, therefore, undertaken to discern the potential role of botulinum toxin therapy in spasticity and in non-motor disorders in adults and children affected by HSP. The novelty of this review is that it represents a comprehensive and critical literature review on this subject, with no other studies of this kind published previously. The current review also includes data not present in previous reviews of this subject [17,18].

## 2. Research Design

We searched Medline and Scopus for papers published up to 1 September 2025. The search terms consisted of botulinum toxin and hereditary spastic paraplegia, as well as botulinum neurotoxin and hereditary spastic paraplegia. Two of the authors independently searched the literature. A third author verified the search results. Excluded from the search were articles in a language other than English, case reports, letters to the editor and reviews. The strengths and weakness of the reported studies, as well as technical issues related to botulinum toxin therapy in HSP, are provided in the Discussion section of this manuscript.

## 3. Results

We found 10 manuscripts that conformed to the search criteria. These articles were published between 2007 and 2024. The results of the search are summarized in Table 1, which includes the authors’ names, date of publication, number of studied patients, type of study, type of toxin used, toxin dose, injected muscles and outcome measures, as well as results and adverse effects (AEs). There were five prospective and four retrospective studies, and one double-blind, placebo-controlled study. One the of ten studies was conducted exclusively in children. The small number (*n* = 12) of children in this study limits the generalizability of the results.

Roussoux et al. [19] enrolled 15 patients (5 women) in a prospective, open label study. The diagnosis of HSP was made based on clinical findings and family history. All patients had severe spasticity and reacted poorly to conventional anti-spasticity medications. Patients were injected with 400 units of onabotulinumtoxinA in different muscles [Table 1]. Gait velocity, MAS and the Patient Satisfaction Scale (0–4) were assessed on the day of injection, and at 2–3 weeks, 2–3 months and 5 months following injection. The authors noted significant tone reduction based on the Ashworth Scale (after adductor injection) and improvement in gait velocity in 8 out of 10 patients following injection of the adductors (*p* < 0.07). All patients reported moderate satisfaction after BoNT treatment.

Hecht et al. [20] reported the results of a retrospective study in 19 patients, with HSP patients receiving onaA and aboA injections in different muscles. The total injected dose was up to 400 units for onaA and up to 1500 units for aboA, respectively. After the injection, MAS improved in 17 of 19 patients, while gait improved in 5 ambulatory patients. All patients reported good or very good treatment experience.

Eva-Dyan et al. [21] conducted a retrospective study of 12 children with HSP. The age of children at the time of injection varied from 2.5 to 15.4 years. Children received onaA (not exceeding 12 u/kg/day) and aboA (not exceeding 25–30 u/kg/day) in different muscles [Table 1]. At one month post-injection, general motor function and quality of motor skills improved in 11 out of 12 (*p* < 0.01) children along with notable improvement in Modified Ashworth grades (*p* < 0.01).

DeNiet et al. [22] prospectively enrolled 15 patients with pure HSP and normal calf strength in a study that investigated the effect of BoNT therapy along with post-injection stretching of the calf muscles on speed of gait and balance. Patients received 500–750 units of abobotulinumtoxinA (Dysport) into the gastrocnemius and soleus muscles. Patients were evaluated first one week before treatment (T0) and at 4 and 18 weeks post-treatment (T1,T2). Patients demonstrated significant improvement in comfortable gait at 4 and 18 weeks after treatment (*p* < 0.01) which was associated with improvement in muscle strength as measured by the Medical Research Council Scale (MRC) (*p* < 0.01) and significant reductions of muscle tone over the same timelines (measured by the Modified Ashworth Scale).

Riccardo et al. [23], in a retrospective study, reported the results of incobotulinumtoxinA (incoA) injections into hip adductors (125 units) and the gastrocnemius (110 units) and soleus (132 units) muscles in 10 patients with HSP. Patients’ responses to BoNT injection were assessed with several scales before injection and at 1, 3, 4 and 5 months after injection. Muscle tone was measured by a device called a myotome. The positive results were as follows: 1—a gradual decrease in muscle tone lasting up to 4 months; 2—a gradual increase in speed of steps, peaking at 5 months, as well as an increase in percentage of back-foot loading.

Servethere and co-workers [24] assessed the efficacy of aboA in 31 adult patients with HSP. Patients were evaluated with several motor and non-motor scales once before and once after the injection. Patients received aboA in different lower-limb muscles (Table 1) with a mean total dose of 1110 ± 535 units. After BoNT injection, they noted significant improvement in tone in the lower-limb muscles, but gait and motor skills did not improve. Among non-motor tests, however, patients’ fatigue improved significantly (*p* = 0.011) (Table 1).

Van Lith et al. [25] prospectively studied 25 patients with pure HSP. Patients were injected with incoA (150–200 units/leg) into the adductors magnus and longus as well as the gracilis muscles. Assessments were performed at T0 (baseline), T1 (6 weeks) and T2 (16 weeks) after treatment. The outcome measures included an assessment of gait (both at comfortable and maximum speed), dynamic balance assessment, muscle strength assessment using MRC and muscle tone assessment (assessed by MAS). At 6 weeks post-injection, both gait width and preferred gait speed increased and improved (*p* = 0.005 and *p* = 0.021) as did lateral balance (lateral stepping). Hip adductor strength decreased at 6 weeks; it gradually returned to baseline at 16 weeks.

In a retrospective report, Parapella and colleagues [26] described the effects of BoNT treatment on 18 adults with HSP. Patients received injections of onaA, aboA and incoA into the proximal and distal leg muscles (Table 1). Along with BoNT treatment, all patients underwent 10 sessions of physiotherapy (each for two hours), which included stretching, postural control, gait training and strengthening of the lower-limb and trunk muscles. Patients’ responses to treatment were assessed by SPRS, MAS, TUG, VAS, the Walking Handicap Scale (WHS), comfortable gait at normal speed and the 10-Meter Walk test (Table 1). Assessments were performed at baseline and at 1 and 3 months post-injection. At one month post-injection, SPRS, MAS, comfortable gait, TUG and 10-meter walking improved significantly compared to baseline (*p* < 0.05). Improvements were maintained and further enhanced at 3 months.

In a randomized, double-blind, placebo-controlled crossover trial, de Lima et al. [27] compared the effect of Prosigne (Chinese neurotoxin) with saline in 55 adult patients with HSP. Prosigne or saline was injected bilaterally into the adductor magnus and triceps surae (gastrocnemius and soleus muscles); the dose of Prosigne was 100 units injected into each muscle. The response of the patients to the injections was assessed with several scales: maximum gait velocity (primary outcome), walking with self-selected velocity, MAS, VAS, MRC (strength assessed on a scale of 0–5), pain severity and pain interference, and the degree of fatigue (assessed by the Modified Fatigue Impact Scale (0–84)). These assessments were conducted first at baseline and then at 8 weeks after each injection (BoNT or saline). The crossover between Prosigne and saline took place between 24 and 28 weeks after the first injection. Except MAS, which assessed reduced muscle tone, none of the scales showed any significant improvement (motor or non-motor).

In a prospective study, Ibrahim and co-workers [28] studied the effect of BoNT therapy (with onaA, incoA and aboA) in 56 patients with HSP. Several muscles in the lower extremities were injected depending on the pattern of spasticity. The most frequently injected muscles were thigh adductors (N = 32), triceps surae (N = 45) and posterior tibialis (N= 20). Patients’ responses to BoNT injections were assessed at one month and three months post-injection with digital gait evaluation, SF12, MAS, SPRS and GAS. At one month post-injection, the gait subset of GAS demonstrated significant improvement in stride time. The SF12 physical component score was also consistent with the improvement in gait parameters.

## 4. Discussion

Our literature search revealed one double-blind, placebo-controlled; five prospective; and four retrospective published studies on the efficacy of BoNT therapy for limb spasticity in HSP. The total number of patients across all studies was 258. All studies, as expected, showed significant reductions in muscle tone after BoNT injections [Table 1]. While all nine non-blinded studies demonstrated degrees of motor and functional improvement (comfortable gait,10-Meter Walk test, lateral balance, gait attainment goal) [Table 1] and patient satisfaction [19,20,22], the blinded study did not demonstrate functional improvement. This discrepancy may be due to the following reasons: 1—The failed blinded study, unlike the non-blinded studies, used Prosigne, the Chinese toxin for injections. Although the authors indicated that one previous study had shown a 1:1 unit ratio between Prosigne and Botox (onaA), it is now well understood that units between marketed toxins are not truly interchangeable [29]. It may be worth exploring novel approaches [30] in future studies that may help interpret results more consistently. Furthermore, the non-blinded studies included in their design combined botulinum toxin therapy with physiotherapy, whereas there was no mention of physiotherapy in the blinded study. The value of concurrent physiotherapy with BoNT therapy in treatment of spasticity has been emphasized by several authors [31,32]. Nonnekes et al. [33], in a letter to the editor, suggested other factors that might explain the failure of this blinded study, such as a lack of inclusion criteria in the design of the study and the use of a low dose of injected botulinum toxin in the muscle by the authors. As can be seen in Table 1, the dose of onaA used for reducing hip adductor spasticity was higher in all non-blinded studies compared to the blinded study (>100 units of Botox) [20,23,27,28]. Furthermore, the difference in injection sites and the fact that in the blinded study, all cases used the same protocol, could also be account for the different results between the open label studies and the blinded study.

Further studies should explore the effect of early injection of BoNT on development of spasticity in patients with HSP. In mice with hereditary spasticity, early injection of BoNTs into spastic muscles prevented development of contracture and allowed development of affected muscles almost to the mature size (within 2%) [34]. In a double-blind, placebo-controlled study of 91 adult patients, Lindsay et al. [35] showed that injection of onaA soon after stroke into the affected muscles slowed down the development of contracture and improved limb function. A recent review of this subject identified 10 studies that stated that early intervention (2–12 weeks post-stroke) with botulinum toxin injection was beneficial to patients with post-stroke spasticity [36].

Our research found only one study in children [21] that pertained to BoNT therapy in HSP. In this small retrospective study, the authors found that injection of onaA or aboA into the hip adductors and the gastrocnemius and hamstring muscles significantly improved general motor function and the quality of motor skills (*p* < 0.05). Although encouraging, the positive findings of this study need confirmation through randomized, blinded clinical trials. Admittedly, these studies would be hard to perform due to the rarity of HSP and the fact that and HSP patients are clinically different and may not respond to the same protocol.

In clinical neurotoxicology, failure to respond to BoNT treatment is often a result of underdosing. In children, selecting the optimal maximum dose of neurotoxin per day is often a challenge due to safety issues. In the study of children with HSH cited in this review, the onaA dose of 12 units/kg caused no serious side effects. Gromley et al. [37], in an open-label study of 438 children with spasticity, reported no serious side effects after repeated injections of onaA with a daily dose of 10 units/kg.

Westhoff et al. [38] reported that injection of BoNT-A into the spastic psoas muscle (under ultrasound guidance and into the thigh) improved gait of two patients with hereditary spastic paraplegia. The iliopsoas muscle is the strongest hip flexor and an important muscle in locomotion. It is worth exploring this option (injecting a single rather than multiple muscles) in children since, if effective, it would save children multiple injections and would result in a lower total toxin dose per injection session.

Bladder dysfunction occurs in 72–77% of patients with HSP [39,40]. Spastic bladder in patients with HSP, which occurred in 82.7% of patients, was associated with detrusor muscle hyperactivity [40]. The most common symptoms were urinary urgency (72%), frequency (65%) and incontinence (55%). Botulinum neurotoxin injections into the detrusor muscle of the bladder are now an approved and extensively practiced approach for management of bladder symptoms caused by detrusor hyperactivity [41]. We found no reports of any prospective clinical trials treating bladder dysfunction in HSP patients with injection of BoNT into the bladder wall. One retrospective report of 71 German patients with HSP and bladder dysfunction included three patients who received BoNT injection into the detrusor muscle; two patients reported subjective improvement in urinary symptoms [42].

Several non-motor symptoms, such as spasticity-associated pain, fatigue, insomnia and depression, can impair quality of life in HSP patients. In one study, 72% of patients with HSP complained of back and limb pain [8]. In another study, pain severity correlated with maximum walking distance and with activities of daily living [9]. Two studies in this review reported improvement in local muscle pain and painful muscle spasms following BoNT injection (Table 1 [20,22]). Significant improvement in spasticity-associated pain has been reported after botulinum toxin injection into the spastic muscles of patients with post-stroke spasticity [43,44]. The analgesic effects of BoNTs after intramuscular injection are attributed partly to reductions in muscle stiffness and spasms and partly to their inhibitory effect on the known pain transmitters (glutamate, substance P and calcitonin gene-related peptide) [45,46].

Depression is common among patients affected by HSP. In a study of 48 adults with HSP, authors found a prevalence of 56% for depression (13% moderate, 2% severe) [47]. Several blinded studies have reported significant improvement in depression after injection of onabotulinumtoxinA into the glabellar and frontalis muscles [48]. Servelhere et al. [24] (Table 1) found significant improvement in fatigue after BoNT injections into the muscles of patients with HSP, but no improvement in depression. However, the role of BoNT therapy has not been investigated for treatment of depression in patients with HSP through blinded studies.

## 5. Conclusions

To our knowledge, this is the first comprehensive review of botulinum toxin therapy in hereditary spastic paraplegia. The limitations of this review are the exclusion of reports in languages other than English, the small number of studied cohorts and the inclusion of retrospective studies (40%), which biases the results. There is a need for double-blind, placebo-controlled studies (preferably with FDA-approved neurotoxins) in both adults and children, and in larger cohorts, to discern the role of BoNT therapy in HSP. There is also a need to determine the value of early BoNT treatment in children affected by HSP. Such studies should employ BoNT doses that are found to be safe for treating children.

## Figures and Tables

**Table 1 toxins-17-00503-t001:** Studies on botulinum toxin treatment of hereditary spastic paraplegia.

Authors and Date	#pts	Study Type	Toxin Type	Total Dose (u)	Muscles Injected	Primary Outcome	Results	Adverse Effects
Roussoux et al., 2007 [19]	15	Pros	onaA	400	Adductor magnus, soleus, gastrocnemius, flexor digitorum longus	Gait velocity, Modified Ashworth Scale (MAS), Patient Satisfaction Scale (0–4 scale)	Gait velocity improved in 8 out of 10 patients following injection of hip adductors; all injected patients were moderately satisfied	Two patients reported local pain at the site of injection for two days
Hecht et al., 2008 [20]	19	Retro	1. onaA 2. aboA	1. Up to 400 U 2. Up to 1500 U	Psoas, hip adductor	Modified Ashworth Scale, patient subjective rating of 0–3, gait assessment	Gait improved in 5 out of 12 patients; all patients reported good or very good response (2 and 3); MAS improved in 17 out of 19 patients; muscle spasms improved	Reversible weakness: 4; transient local pain: 1
Geva-Dyan et al., 2010 Children [21]	12	Retro	onA, aboA	Total doses did not exceed 12 u/kg (onaA) and 25–30 u/kg (aboA) Dose varied per muscle	Gastrocnemius, adductor, hamstring	Modified Ashworth Scale, general motor function measure, quality of motor skills	General motor function improved (11/12: *p* 0.01) Quality of motor skill improved 10/12: (*p* 0.01)	Transient weakness: 1; transient local pain: 1
De Niet et al., 2015 [22]	15	Pros	aboA	500–750 u	Gastrocnemius, triceps surae	Comfortable gait velocity, maximum gait velocity, Modified Ashworth Scale, muscle strength measured by MRC	Comfortable gait velocity increased by 9% and 12% (*p* < 0.05); 12 out of 15 patients expressed satisfaction; in two patients, muscle spasms improved	Transient weakness: 3
Riccardo et al., 2016 [23]	10	Retro	incoA	Mean doses: hip adductor: 125 u; gastrocnemius: 110 u; soleus: 132 u All injected bilaterally	Adductors, gastrocnemius, soleus	Speed of step, foot pressure, Modified Ashworth Scale	Speed of step showed a gradual increase, peaking at 5 months	Not mentioned
Servelhere et al., 2018 [24]	33	Pros	aboA	1110 ± 535 u	Adductors, hamstring, soleus, gastrocnemius, tibialis posterior, quadriceps	Modified Fatigue Impact Scale (MFIS), Modified Ashworth Scale, gait velocity, 10-Meter Walk Test	MFIS improved (*p*: 0.011); adductor tone improved (*p* < 0.05); no improvement in 10-Meter Walk Test	Transient lower-limb weakness: 1; increased sleepiness: 1
Van Lith et al., 2019 [25]	25	Pros	incoA	150–200/leg	Hip adductors, gracilis muscle	Gait width, quality of sideways stepping, gait speed, Modified Ashworth Scale	Preferred gait speed and lateral balance improved significantly	None
Paparella et al., 2020 [26]	18	Retro	onaA, incoA, aboA	Dose was determined based on the patients’ weight	Adductors, hamstring, soleus, rectus femoris, gastrocnemius	Comfortable gait velocity, SPRS, TUG test, VAS, NRS, Modified Ashworth Scale	Significant improvement in VAS, NRS, SPRS, gait velocity, TUG test and modified Ashworth Scale (*p* < 0.05)	None
de Lima et al., 2021 [27]	55	DB-PC	Prosigne versus saline	400 u: 100 units into each muscle bilaterally	Adductor magnus, triceps surae	Comfortable and maximal gait velocity, SPRS, Modified Ashworth Scale	Adductor muscle tone decreased in Prosigne group (*p* = 0.01); no significant difference between two groups regarding gait or SPRS	Side effects were noted in 14% of toxin group and 7% of saline group; all mild and transient
Ibrahim et al., 2024 [28]	56	Pros	onaA, incoA, aboA,	Mean doses: adductor: 133 u; triceps surae: 109 u; tibialis posterior: 64 u	Adductors, tibialis posterior, triceps surae, flexor digitorum brevis, rectus femoris, biceps femoris	SPRS, Modified Ashworth Scale, Patient Goal Attainment Scale (GAS): 0–4	One month post-injection, SPRS, stride velocity and Ashworth score significantly improved (*p* < 0.5); GAS improved in 66% of patients	Not mentioned

#pts: number of studied patients; DB-PC: double-blind, placebo-controlled; Pros: prospective; Retro: retrospective; SPRS: spastic paralysis rating scale; TUG: time up and go; GAS: Goal Attainment Scale; VAS: visual analog scale (pain and quality of life); NRS: numerical rating scale (pain); MRC: Medical Research Council Scale.

## Data Availability

No new data were created or analyzed in this study.

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
