# Peer review of "Botulinum Toxin Treatment in Hereditary Spastic Paraplegia—A Comprehensive Review and Update"

_toxins, 2025, doi:10.3390/toxins17100503_

Round 1

Reviewer 1 Report

Comments and Suggestions for Authors

The authors offer here a review of literature on the use of toxin botulinum (TB) in patients with HSP.

About the rational of this work : botulinum toxin is already quite the reference treatment in all disorders that generate spasticity, including HSP. However, providing proof of efficiency or criticizing ancient procedures is always an interesting approach.

General comments :

  • The introduction is well written and sums up the key features that readers may want to know about HSP. However, spasticity and its management is barely discussed and consequently, this review is not “justified”. I believe the authors should spend more time on the discussion of why it is necessary to review the use of toxin botulinium, because right now the topic of the article is almost dissociated from the introduction.
  • The result section is very interesting however some of the data displayed are not necessarily relevant to the point of the review (use of TB in HSP) such as giving how many were autosomal dominant
  • the difference in injection sites and the fact that patients all had the same protocol in the double blind study should also be discussed as explanations for the discrepancies in efficiency observed with open studies
  • The first paragraph of the discussion should be in the introduction alongside with an explanation of why this review is needed or relevant
  • The authors mention several times that the effect of toxin botilinium in HSP needs to be proved through more robust studies, with different designs, double-blind etc. However, they should also mention why such studies are difficult if not impossible to perform : rarity of patients, the fact that all patients are clinically different and cannot respond to a same protocol, and so on

Author Response

Reviewer 1-

Thank you for constructive comments and suggestions. We have revised the manuscript according to your suggestions. All changes are marked in red on the marked copy of the manuscript. 

General comments:

  • The introduction is well written and sums up the key features that readers may want to know about HSP. However, spasticity and its management is barely discussed and consequently, this review is not “justified”. I believe the authors should spend more time on the discussion of why it is necessary to review the use of toxin botulinium, because right now the topic of the article is almost dissociated from the introduction.
  • Answer: Following your advice, we expanded the introduction to include statements on justification of this review (lines 97-106 of revised manuscript).
  • The result section is very interesting however some of the data displayed are not necessarily relevant to the point of the review (use of TB in HSP) such as giving how many were autosomal dominant
  • Answer: Following your recommendation, we took the genetic information out of the manuscript in the result section and shortened this section. Please see lines 129 to 211 0f the revised manuscript.

  • The difference in injection sites and the fact that patients all had the same protocol in the double- blind study should also be discussed as explanations for the discrepancies in efficiency observed with open studies
  • Answer: Per your advice a statement to this effect was added to the discussion (lines 234 to 236, revised manuscript)
  • The first paragraph of the discussion should be in the introduction alongside with an explanation of why this review is needed or relevant
  • Answer: Per your recommendation, we moved the first paragraph of discussion to introduction (lines 90-104)
  • The authors mention several times that the effect of toxin botilinium in HSP needs to be proved through more robust studies, with different designs, double-blind etc. However, they should also mention why such studies are difficult if not impossible to perform: rarity of patients, the fact that all patients are clinically different and cannot respond to a same protocol, and so on
  • Answer: We have added your suggested statement to the manuscript (lines 251 to 252).

Thank you again for your time and your helpful comments.

Reviewer 2 Report

Comments and Suggestions for Authors

Dear Authors

I carefully reviewed the review article entitled “Botulinum Toxin Treatment In Hereditary Spastic Paraplegia- A comprehensive Review and Update”. This manuscript addresses an important and underexplored topic: the potential role of botulinum neurotoxin therapy in HSP.

Overall, the review is well-organized, readable, and falls within the scope of Toxins. However, some minor concerns must be addressed to enhance the scientific clarity and presentation of the manuscript. Below are section-wise detailed comments.

Abstract

  • The abstract lacks sufficient detail on inclusion/exclusion criteria; please clarify the methodology.
  • Please include clearly the number of patients included across all studies to provide a stronger context.
  • The novelty claim is weak; highlight how this review adds value compared to prior reviews.

Introduction

  • Please add recent global meta-analysis references.
  • Please discuss pharmacological dosages and treatment limitations.
  • Line 24-26: It is strongly recommended to cite Gan et al. (2024) to highlight how insights from neuroscience and molecular genetics provide broader mechanistic understanding of neurodegenerative disorders such as HSP. This would strengthen the background on heterogeneity and pathophysiological mechanisms (doi: https://doi.org/10.1016/j.cobeha.2024.101431).
  • When discussing the classification of HSP based on genetic and molecular pathways, reference Zhu et al. (2020) should be integrated to illustrate how bioinformatics tools are increasingly being applied to understand complex gene-phenotype relationships in neurological disorders (doi: 10.3389/fbioe.2020.00547).
  • Consider elaborating more on the research background, clearly stating the knowledge gap, the problem being addressed, and how the present review provides the solution in the introduction section.

Results

  • Mention the total number of patients across studies.
  • Line 88-89: Only one pediatric study is mentioned; highlight its small sample size as a limitation here.
  • Line 95–112: Summaries of individual studies are thorough but repetitive; condense while focusing on outcome relevance.

Discussion

  • Line 208-214: When explaining discrepancies between Prosigne and FDA-approved toxins, include Zhang et al. (2025) to highlight the importance of using standardized biologics with validated safety profiles (https://doi.org/10.2147/JIR.S499403).
  • Line 230-240: While discussing pediatric therapy and challenges of optimal dosing, Kang et al. (2024) should be cited to support the idea that immune markers and responses are dynamic, and biologic interventions can have variable effects (https://doi.org/10.1038/s41598-024-75636-2).

Conclusion

  • I strongly recommend separating the conclusion section.
  • Please include a schematic figure summarizing current evidence and gaps.

Author Response

Reviewer 2-

Thank you for constructive comments and helpful suggestions. We have revised the manuscript according to your suggestions. All changes are marked in red on the marked copy of the manuscript. 

Abstract

  • The abstract lacks sufficient detail on inclusion/exclusion criteria; please clarify the methodology.
  • Answer: Per your advice, Inclusion and exclusion criteria were added to the abstract (lines 12 to 15)

  • Include clearly the number of patients included across all studies to provide a stronger context.
  • Answer: Number of studied patients was included in the revised abstract (lines 15 and 16).   

  • The novelty claim is weak; highlight how this review adds value compared to prior reviews.
  • Answer: Following your suggestion a statement on novelty claim was added to the abstract (Lines-26-28).

Introduction

  • Please add recent global meta-analysis references.

             Answer: Provided in Line 105-106

  • Please discuss pharmacological dosages and treatment limitations.
  • Answer: Per your advice, pharmacological dosages and treatment limitation were added to the introduction (Lines 74-79).
  • Line 24-26: It is strongly recommended to cite Gan et al. (2024) to highlight how insights from neuroscience and molecular genetics provide broader mechanistic understanding of neurodegenerative disorders such as HSP. This would strengthen the background on heterogeneity and pathophysiological mechanisms (doi: https://doi.org/10.1016/j.cobeha.2024.101431).
  • Answer: Added per your advice: Please see lines 53-54 of the revised manuscript.

  • When discussing the classification of HSP based on genetic and molecular pathways, reference Zhu et al. (2020) should be integrated to illustrate how bioinformatics tools are increasingly being applied to understand complex gene-phenotype relationships in neurological disorders (doi: 10.3389/fbioe.2020.00547).
  • Answer: Added per your advice - Lines 55-56 of the revised manuscript

  • Consider elaborating more on the research background, clearly stating the knowledge gap, the problem being addressed, and how the present review provides the solution in the introduction section.
  • Answer: Lines 97-106 of the revised manuscript

Results

  • Mention the total number of patients across studies.
  • Answer: Line 215 of the revised manuscript

  • Line 88-89: Only one pediatric study is mentioned; highlight its small sample size as a limitation here.
  • Answer: Please see line 124of the revised manuscript

  • Line 95–112: Summaries of individual studies are thorough but repetitive; condense while focusing on outcome relevance.
  • Answer: Description of individual studies is shortened and focused more on evaluation and outcome. Lines 129-211

Discussion

  • Line 208-214: When explaining discrepancies between Prosigne and FDA-approved toxins, include Zhang et al. (2025) to highlight the importance of using standardized biologics with validated safety profiles (https://doi.org/10.2147/JIR.S499403).

             Answer:  Cited per your advice – please see lines 224-225

  • Line 230-240: While discussing pediatric therapy and challenges of optimal dosing, Kang et al. (2024) should be cited to support the idea that immune markers and responses are dynamic, and biologic interventions can have variable effects (https://doi.org/10.1038/s41598-024-75636-2).
  •  

             Answer: The manuscript cited below was retrieved from information you kindly provided us (above).  Is this the correct article?  Please advise how the information from this article relates to botulinum toxin therapy in pediatric patients.  

Kang, S., Wu, Q., Shen, J. et al. CD27 is not an ideal marker for human memory B cells and can be   modulated by IL-21 upon stimulated by Anti-CD40. Sci Rep 14, 23742 (2024). https://doi.org/10.1038/s41598-024-75636-2

Conclusion

  • I strongly recommend separating the conclusion section.

             Answer: separated per your advice

  • Please include a schematic figure summarizing current evidence and gaps

             Answer: Provided.

Round 2

Reviewer 2 Report

Comments and Suggestions for Authors

I have carefully reviewed the revised version of the manuscript. The authors have made significant revisions and addressed all of my previously raised comments. I am fully confident that the article can be published in its current form.